# The Meaning of 'Spiritual' as Integral Health: From Hippocrates of Kos to the Potamius of Lisbon

Alex Villas Boas * and Isidro Lamelas 

CITER—Research Centre for Theology and Religious Studies, Faculty of Theology, Universidade Católica Portuguesa, 1649-023 Lisbon, Portugal
* Correspondence: alexvboas@ucp.pt

**Abstract:** This article aims to analyze how the category 'spiritual' used by Hippocrates of Kos can help with a better understanding of the influence and reception of Hippocratic medicine the Christian self-understanding as a religion of healing, especially from the Hippocratic influence in Potamius of Lisbon, and at the same time this Christian understanding contributes to the desacralization of medicine as a medical art. For this purpose, it will be analyzed the category pneuma in the Hippocratic naturalism, and within the debate between the medical schools, Pneumatics and Empirics, around the various methods of treatment to maintain the dynamization of pneuma. With this, it is intended, then, to identify different forms of reception of Hippocrates in Christianity associated with the different perceptions that one has of the writings of the physician of Kos. Such contextualization aims to help understand the process of spiritualization of pneuma that paved the way for the radicalization of the Pauline duality between body and soul, as well as to identify another understanding of pneuma linked to the conception of stoic *sympatheia* and the reading of the empiricists of Hippocratic Naturalism, both present in the Christian reading of the *Corpus Hippocraticum*. In this sense, this article will take as an example the work of Potamius of Lisbon (4th century), in order to identify an epistemological model of spirituality and health that could works as a kind of antidote to the tendency towards spiritualization of the pneuma, to accentuate its aspect of integrating, vitalizing and unifying body and soul in a pneuma dynamism, connecting the notion of restoring the health of nature with the notion of Christian redemption.

**Keywords:** spiritual; *pneuma*; Hippocrates of Kos; Potamius of Lisbon; spirituality and health; integral health; ancient christian thought

## 1. Introduction

In its most common sense, the Latin term *spiritus* is translated from its Greek counterpart *pneuma* (πνεῦμα), which is taken from the verb *pnéo*, used for the action of blowing, whether linked to breathing or the movement of the winds, or even to blow a musical instrument, and can be related to the exhalation of odors. However, the presence in Greek of other words to refer to atmospheric air (αέρ), the exhalation of odors (ατμός), and even to the wind (άνεμος) suggests the need to better understand the specific use of the term in question. It has a broad and dense semantics that has developed over centuries in the West, covering various fields such as religion, theology, physics, philosophy, and Hippocratic medicine.

What matters here is to capture how the term *pneuma* developed conceptually in the Hippocratic tradition and how such medical semantics were incorporated into early Christian theological thought. It is worth mentioning that *pneuma* was not used in the religious sense in classical Greek literature, which tends to use the term *psyché*. Its religious incorporation takes place with the Greek translation of the Hebrew Scriptures carried out in Alexandria in the 3rd century BC, which became known as the Septuagint or LXX Translation, evoking the legend of the number of translators who were separated and

through divine inspiration produced the same version (Dines 2004). In the LXX, the feminine word *Ruah*, used to refer to God's action, is translated as *Pneuma*, of neuter gender, which in turn will be translated into Latin as *Spiritus*, of masculine gender, in Jerome's Vulgate, in the 5th century AD, following the custom, above all, of Cicero.

That said, the Hippocratic use of the term is fundamental to determining the discursive context of its meaning and its incorporation into Greco-Roman philosophical and Christian theological thought. The Hellenic context of its development is decisive for the interpretation of the Pauline duality of *body-soul* (σάρξ-πνεῦμα/*corpus-spiritus*), at times accentuating the spiritualization that tends to depreciate the body, at other times accentuating a fundamental nexus between the two elements (Amundsen 1982, p. 345). The choice of analyzing Potamius of Lisbon aims to identify, in its rhetoric, the elements of Hippocratic naturalism that constitute its anthropology and dogmatics, a path less explored due to the strong influence of Platonic currents in the West, but with different consequences for the dialogue between religion and health, which here we intend to rescue and explore.

## 2. *Pneuma* in Hippocratic Naturalism

The emergence of Hippocratic medicine takes place within the Asclepiades Temples and is accompanied by the expansion of the cult of Asclepius that lasts for eight centuries, from the 4th century BC to the 4th century AD, thus entering the Roman Empire and reaching Imperial Christianity. Hippocrates himself is an asclepiad, a member of this peculiar religious community constituted by a kind of physician "priests" (Plato 1921, *Pheadrus* 270c; *Protagoras* 311b). To the numerous shrines of this god *soter*, regarded as the great "friend of men" (φιλάνθρωπος), pilgrims flocked from everywhere in search of healing of the soul and body (Villas Boas 2020).

However, despite the close relationship between Hippocrates and the cult of Asclepius, one of the gods to which the medical oath is made, its medical concept can be understood within what will be called *Hippocratic naturalism* here. In general terms, the so-called Ionian science would have influenced Hippocrates of Kos to think of Asclepiad healing practices in a new way. Especially the theory of atoms would have influenced the idea of the four humors: blood, phlegm, yellow, and black bile (Gramaticos and Diamantis 2008, p. 3) and its relationship to the theory of the four elements (cold, dry, hot, liquid), related to Empedocles' (494–434 BC) four elements (fire, water, air, earth). The main question is that pre-Socratic philosophers' Ionian science opens up a new way of observing nature (*theoria*) which guides Hippocrates' analysis in looking for the natural causes of disease (Villas Boas 2020). Furthermore, the medical art based on cooperation between doctors and patients in the face of illness, without direct relationship with the gods, is supported in this belief in nature (*Corpus Hippocraticum* [CH], *Epidemics*, I, 11; LCL 147)[1].

The use of *pneuma* in Hippocratic medicine itself can be better understood in this naturalistic context. Such Hippocratic naturalism can be described in a system that consists of the body with three organs (the heart, lungs, and brain) and its main agents, congenital heat and *pneuma*. They are present specifically in the body, identified in the air breathed, which, through the lungs, is passed into the blood. It can also be found in gases exhaled by the body. This composition thus constitutes a mechanical physiology that maintains the harmonious and balanced unity of the body parts and their liquids or the four humors as a fundamental requirement for the health of the body and the *psyché*. Once heat keeps the body alive residing in the left ventricle of the heart, the *pneuma* (breathed air and blood) feeds, drives, refreshes, and enlivens the body functioning and consequently allowing the *psyche*'s balance of humors. The air-*pneuma* enters the mouth, travels to the nose and ascends to the head. From the head, it descends to the belly, passes through the lungs and arteries and, from these, spreads throughout the body (*Sacred Disease*, 4; LCL 148). In the heart and lungs, the *pneuma*, mixed with the blood, feeds the generated heat while refreshing the body's parts. From the brain and lungs, *pneuma* travels through blood vessels to different parts of the body, ensuring its vitality and movement. *Pneuma* is thus vitality itself or even a vital force (*dynameis*) for the functioning of the body (Jouanna 12). The

pathological state would result from the lack, excess, or isolation of any of these liquid-humors from any part of the body. Treatment basically consisted of restoring the good crisis, that is, the balanced mixture of strength and quantity of these humors. Hippocratic nature can be defined essentially as a relationship between the body and the *pneuma* that moves it in order to generate harmonious functioning in the four constituent elements of the organism (*On Diseases*, 32–33; LCL 472; *On the Nature of Man*, 4; LCL 150). In the book dedicated to epilepsy (*Sacred Disease* 1, 1), until then understood as a sacred disease, Hippocrates speaks of it as a disease like all others, therefore of natural causes, and in particular a hereditary problem in the circulation of the veins, which complicates the passage from *pneuma* to the brain (*Sacred Disease*, 5, 1–4). In the pneumatic vitality of Hippocratic naturalism, one can identify an influence of (or convergence with) the pre-Socratic Anaximenes of Miletus (588–524 BC), of seeing the *pneuma* as the *arché*, or principle of life (Aristotle 1984, cf. *Metaphysics*, 11, 1066b), being the most indispensable and most important element for health, as different diseases originate from its variations in the body (Nutton 2018, pp. 66–67).

The emergence of medical schools, in general, is not based on Hippocratic naturalism, but rather on the debate around various methods of treatment (*diversas curandi vias*, cf. *De Medicina*, Proemium, 8) able to maintain the balance of health through the dynamization of *pneuma*, namely, through diet (*diaitiké*), medication (*pharmakeutiké*), and hand technique or surgery (*keirourgia*), as later indicated by the Roman encyclopedist Cornelius Celsus (1935, *De Medicina*, Proemium, 9).

### 3. *Salus'* Ideal in the Pneumatic Medical School

The first medical school of antiquity was founded by Thessalo, son of Hippocrates, and Polybius, his son-in-law, in the 4th century BC, and continued by Diocles of Carystus, Praxagoras of Cos, and Chrysipo, who were considered the first students of the doctor of Cos. They became known as the *Dogmatic School* for professing the theoretical opinions of Hippocrates, that is, the *dogmas*, as the philosophical vocabulary was understood at the time. This is why they are also sometimes called *Hippocratici* (Smith and Anthon 1900, p. 372; Celsus 1935, *De Medicina*, Proemium, 8).

*Dogma* or *dogmata* in the 1st–2nd century AD, can be understood to mean beliefs or opinions that are evaluated according to the action they promote, and in this sense, it is related to discernment or *prohairésis*, and must preserve the good or correct *dogma*, to be understood as adequate to reason and to eliminate the bad or false *dogmas*, those contrary to reason, which bring about difficult and painful situations by supporting disturbing images of reality [*tracheia phantasia*] (Dinucci 2017, pp. 101–4).

In this sense, the dogmas of medical schools concern the *great debates* (*magnas contentionis*) about the fundamentals and practices of medicine. Such discussions, or dogmatic debate, unlike what could be inferred by contemporary common understanding, not only remain open to considering new opinions (*addicta opinioni*), as it is not necessary for everyone to be in total agreement, nor are they excessively divergent, but they are intended to be a kind of intermediary place (*media inter diversas sententias*) that allows the search for the truth (Celsus 1935, *De Medicina*, Proemium, 45).

However, the dogmatic medical school in general claimed the need for a knowledge of nature (*naturae cognitionem*) as a condition to better conduct treatment (Celsus 1935, *De Medicina*, Proemium, 10). Dogmatists then professed an epistemology based on a grounded theory of medicine (*rationalem medicinam*) that was guided by the formulation of principles (*principis*) in which knowledge of the *hidden causes* of the disease would first be necessary (*abditarum*), which often had to do with the patient's way of life and history and their practices that resulted in excesses or shortages, to then correlate with the *evident causes* (*evidentium*), and therefore with natural actions (*naturalium actionum*) and, finally, the organs or internal parts (*partium interiorum*). The reason why one should begin by investigating the hidden causes is due to the frequent manifestation of new diseases that the usual practices

do not reveal, thus demanding to know how the genesis of the disease took place (*De medicina*, Proemium, 13–17).

From this dogmatic medical school there is a variant in the 1st century AD called *pneumatikoi* or *Pneumatic School* formed by Atheneus from the province of Cilicia, from the city of Attalia, or from Tarsus, the provincial capital, according to Caelius Aurelianus (1950, *De morbis acutis et chronicis* II, 6). Attila of Athenaeum, then, influenced by Posidonius Stoicism (Nutton 2018, p. 203), begins to interpret the pre-Socratic and Hippocratic pneuma as the Stoics, which identifies the soul with the *pneuma* (*Stoicorum Veterum Fragmenta* 1964, [SVF], pp. 72, 135, 138), according to Zeno, in which the soul is a psychic *pneuma* described as "a hot *pneuma* that penetrates and moves our entire being" (SVF I,135). Augustine had already noticed this particularity, when he recalled that "for Stoicism the soul is nothing but the breath [*spiritus*]" (Augustinus, *De Civitate Dei*, XIV,2). In other words, the *psychic pneuma* is "that by which its output determines death" (SVF I,136; 368; 443/2; 796).

Furthermore, just as the *pneuma* allows the connection of the whole body, the pneumatics shared with the Stoics the hypothesis (dogma) that the entire Cosmos, and therefore all things, were interlinked by *connections* (*sympatheia*) generated by the same principle, that is, the *pneuma*. So, the reason (*logos*) would have the purpose of observing nature, to provide the principles for a salutology, an understanding of health (*salus*) that would involve the balance between the parties and between the *physis* of the person and that of the *cosmos*. For that, the practical experience of the physician is not sufficient. In this sense, medicine emphasizes the need to incorporate the wisdom of philosophers (*ex sapientiae*, cf. *De medicina*, Proemium, 14) and it engenders an *ascesis* or a *spirituality of health* that seeks to capture the *logos* that unveils the vital *pneuma* that dynamizes the Cosmos, to live in harmony with it, since the same *pneuma* dynamizes human life. Spirituality is understood as *self-care exercises*, which include health care in harmony with the nature of the whole, with the human being understood as a micro-cosmos or *imago mundi*. Such spirituality thus implies simultaneously a knowledge of oneself and knowledge of the world through spiritual exercises that aim to learn to live by dedicating oneself to the practices of self-care which includes the love of wisdom, through meditation on good things, reading philosophers and poets, research and dialogue, reading poems, abandoning the superfluous, therapy of passions, carrying out duties, mastering oneself, and indifference to what is not essential for living well. In short, the spiritual exercises of the Stoics aimed at an awakened awareness and lucidity so that as you learn to live, you also learn to die. The "exercise of death" was specifically linked to the passage from the contemplation of individuality to the contemplation of universality, and thus living and dying in harmony with nature (Hadot 2014, pp. 22–35, 44–50).

## 4. The Empirical School's Reaction

For Serapion, Apollonius and Glaucias, the pneumatics distance themselves from what the medical art should be, and therefore, a knowledge based on "practice and experience" (*usu et experimentis*), where the *logos* of medicine is based on the *physis* of the body (*De Medicina*, Proemium, 10). This is the reason why Serapion and others created the School of Alexandria, or more precisely, the Empirical School.

However, among the discussion of whether the *logos* of medicine should be guided by the *physis* of the body or the *pneuma of the cosmos*, there are those who postulate the need for observations based on what diseases have in common, which are a sufficient knowledge for the treatment to be adopted. Because they believe the *method* is a path to the cure, it does not require knowing the cause of the disease or the patient's history, and thus, they will classify themselves as *Methodists* (*De Medicina*, Proemium, 57).

Given the variation in the positions of each school, Archigenes tries to create something that combines the strengths of each one, and thus founds the *Eclectic School*. However, the eclectic ones seem to return to religious practices in treatment, especially in their pharmaceuticals, in the case of "antipathetic medicines", which suggested a symbolic movement contrary to the disease, to produce symptoms opposite (*anti*) to the patient's and

which permitted, for example, the use of amulets (Temkin 124). Such an eclectic posture is certainly not a problem for the medicine exercised in the Asclepiades Temples and, despite maintaining the diagnosis of the natural causes of diseases and adopting religious practices in the treatment prognosis in line with the religious culture of a "supplicant piety", this reinforced the idea of deifying the forces and agents of nature. Although, it is quite contrary to the stoic "reflective piety", which incorporates pre-Socratic physics, and proposes the ideal of *salus* nature and its rules as a norm of healthy life. Even these different types of Greek piety related to health or, one might say, Greek health spirituality that make up the *salus* ideal are considered acceptable from a medical point of view. For the other schools, everything follows the natural order, so that every disease result from a change in nature or from something that went "against nature" (παρα ουσιν); healing consists of putting back the order of nature [κατα ουσιν] (CH, *Epidemias*, VI,5,1 (230)).

**5. Hippocratic Reception in Christianity**

Dissociating medicine from the Asclepiad cult, the Hippocratic naturalistic epistemology and the religiosity of healing, it is possible to better analyze the process of receiving Hippocrates in Christianity. Another element to be observed is that the existence of the multiplicity of medical schools is not unknown to Christians of the first centuries (Clement of Alexandria, *Stromata* VII,15,90,4), nor a reason to refuse medicine, but rather allowed them to know the themes of Hippocratic medicine, not necessarily through Hippocratic sources or through direct contact with medical science, but through *eruditio veterum* of the Hippocratic Tradition marked by religious and philosophical eclecticism, especially through Stoicism and Medioplatonism that, to a large extent, wanted to respond to the search for *salus* that characterized the image of man and the world in which Christianity emerged (Von Harnack 1906, pp. 104–8). The reception of Hippocrates in Christianity, therefore, must consider the debate on the interpretation of the *Corpus Hipocraticum* in the emergence of new Hellenic and Roman medical schools in the 1st century AD.

*5.1. The Salvation of the Deus Clinicus and the Religion of Healing*

The emergence of epidemic diseases in an environment of increasing health precariousness due to the context of constant internal and external wars waged by the Roman Empire coincided with the same conditions of epidemics and war in which Hippocratic medicine developed in classical Greece (Villas Boas 2020). This proximity of context favored the expansion of the Asclepiad temples by the emperors, in which the Hippocratic medicine was practiced. With this, Asclepius, the *theós soter* of Greece also becomes the Roman *Aesculapius*[2], bringing with him the medical skills of his priests.

The *soter* epithet to which Asclepius came to be invoked gained a new theological semantics in which the virtue of healing became central, in such a way that even Zeus and Apollo also began to receive the same invocation of *soter* (Tatian, *Discourse to the Greeks* 8), as well as the cult of Hygeia and Panacea are incorporated in the Asclepiades temples, as stated in the Hippocratic oath (*Orkhós* 1). In this context of a religious culture of healing, the message of redemption from Christianity in the Greek world was received in the form of the *Gospel of the Savior and of healing*[3], with Jesus Christ being understood as the "physician" who addresses sick humanity and promises him health, and the Gospel of Luke, a "physician", the one that would have been better received and would have opened the door to the other evangelists in Greco-Roman culture. Such an encounter between Asclepius and Jesus Christ was marked by the dispute over who was the true savior, the true *deus clinicus* (Von Harnack 1892, pp. 2–5, 89–90; 1906, pp. 91–92; Dal Covolo and Gasparro 2008).

Oriental healing cults can also be identified, such as the Egyptian deity Isis, which had been amalgamated with Greek religiosity in the cult of Serapis since the Hellenistic period when the Macedonian general, Ptolemy I, became Pharaoh of Egypt, with the death of his friend, Alexander the Great (Plutarch 1939, *Moralia* V 27–29). Amid the great demand for healing and the different offers of healing religion there are a number of contrasting

attitudes from different social strata. There are people who see medicine as a knowledge that looks for diseases in their natural causes; there are authors who see these natural causes as close causes of an ultimate supernatural cause; and just as those who see disease in divine causes also seek religious solutions. There are also thinkers who see health within more radical dichotomies, whether theoretically, whether in their ascetic practices, and there are thinkers who tend to see the relationship between body and soul in a healthy way and recognize the value of the medicine; but also, there are those who, by demonizing Hellenic culture, also attack the medical art. This plurality of attitudes and views on medicine can be found both in the so-called Pagans and in Early Christianity (Amundsen 326–50).

Nevertheless, the configuration of a subjectivity marked by the development of an urban culture with its dynamics of individualization of relationships, ended up generating greater awareness and concern for personal suffering, and the more educated classes tend to accept medical practice as a curative part. Even so, there are a number of patristic sources on which some Christians still condemn medical practice, perhaps because the asclepiad physicians swore an oath to healing deities, or because there was a practice of the patient to consecrate his life to these same deities (Von Harnack 1906, p. 92), or even because, for the Roman interpretation of Hippocrates, medical practice is not dissociated from a divinatory art, like some kind of mystical experience that produces some rational insights, as it is the case of Galean (Van Nuffelen 2014, pp. 337–52).

That said, it is worth noting that Christian authors are not in agreement regarding traditional medical art, so there is an ambivalent attitude in Christianity of acceptance and rejection of Hippocrates, which is mainly due to the sacralizing tradition of medicine and its main actors, the asclepiads, the eclectic school, and the interpretation given to medicine by Galean. Despite the naturalistic diagnosis of diseases that were dissociated from divine causes, as Hippocratic medicine developed within the Asclepiad shrines and the cult of Asclepius, obtaining a cure was not dissociated from the belief that the pagan deity healed. The historian Pausanias writes that it was a custom in the 2nd century AD for the Asclepiads to elaborate their genealogies to assume themselves as "sons of Asclepius", continuing to claim the divine character recognized by Asclepius (*Description of Greece,* Ill, 26, 9[4]; Temkin 181).

In this sense, the rejection of Hippocrates is linked not properly to Hippocratic medicine, but to the context of dispute over what was the *vera religio* through the offering of healing, given the crisis in the belief of the Olympic gods dating from the previous centuries. Thus, the criticisms made of the practices of the Olympic religion focused on the Asclepiades as their origins date back to the Olympic affiliation, since Asclepius is considered a friend of Apollo. As early as the 2nd century AD, and therefore, five centuries after the time of Hippocrates, one of the greatest criticisms leveled at this "medical god" it wasn't exactly about the ability of their medical practices to achieve healing for patients, but it was, in fact, the characteristic of "liking money", at least of some asclepiads (Clement of Alexandria, *Protrepticus* 2, 30.1; *Sources Chrétiennes* 1941, pp. 2, 85; Arnobius, *Adversus nationes* IV,25; *Biblioteca de Autores Cristianos* 1954, pp. 238–239, 632; Temkin 1991, pp. 98–105) and thus of only deceiving those who ignore the true God (Clement of Alexandria, *Protrepticus* 2, 26,7; SC 2, 82). Tertullian confirms the "greed", recalling that "Aesculapius had been struck by lightning, in punishment for his greed, criminally exercising medicine" (*Apologeticum* 14,5; CCL 1, 113). In other words, the criticism of the first Christian apologists concerns the deification of physicians, who, due to the importance of great historical characters, were commonly "venerated as gods", as it was the case with Galen, according to Eusebius of Caesarea (*Historia Ecclesiastica* V, 28, 14; SC 41, 78). And in this context, medicine ended up being connoted with a belief in occult powers, so that Tatian in this scenario even questioned whether "the art of medication (*pharmakeutiké*) was a kind of the same trick" (Tatian, *Discourse to the Greeks*, 18; BAC p. 598). Also, he has refuted the doctrine of "sympathy" defended by Democritus, ruling that no disease can be expelled through "antipathy", which is confused with a magical approach to therapy (Tatian, *Discourse to the Greeks*, 17; BAC pp. 596–597; Temkin 1991, p. 124).

It is worth mentioning that Tatian, Tertullian and Arnobius are part of a less representative tradition within Christianity that adopts a position of rejection of Hellenism in the face of the "wisdom of the world" so often connoted with a sacral vision of knowledge and perspectives that "deify the elements of the world", and therefore, they fight, not against the medicine as such, but against the sacred or "demonic" use of pharmaceuticals (Tatian, *Discourse to the Greeks* 18; BAC p. 598; Tertullian, *Apologeticum* XXII, 11; CC 1, 124). But even Tatian from Syria, the most radical, when defending a separation between Christianity and the "wisdom of the world", and therefore considering medicine deceit and an illusion of the devil, he did not exclude, at the same time, *dietetics* or *surgery* despite the priority of *salus animae* over bodily health (Cf. Tatian, *Discourse to the Greeks*, 17; 596–597; Cyprian, *De mortalitate*, 91; BAC 717, 292; Augustinus, *De civitate Dei*, V, 2).

Origen also responds to Celsus, who accused Christians of "avoiding doctors", asserting that "we [Christians] do not run away from doctors" and believing that "true doctors in no way corrupt those they promise to treat" (Origen, *Against Celsus*, III, 75; SC 136, 168). The same Alexandrian sage even argues that we must know the method and principles of medicine, to know how to act and intervene in the disease, to establish a measured and demanding diet, to measure the heat of fever in the beat of the pulse, to dry out the abundant humors as periodic cures, and to moderate and contain them. If someone just knows these things and doesn't put them into practice, their science is useless and vain (Origen, *In Lucam homiliae*, 1; cf. CH, *On decency*, 1, 95–96; LCL 148).

Regarding ancient Christian authors then, even the most reticent in relation to the pagan wisdom of the Greeks, it must be taken into account that everyone recognizes that medicine, "like all science", "comes from God, as a science of health in which one learns to know the virtues of plants, the qualities of their juices and the difference in their effects" (Origen, *In Numeros Homiliae* XVIII, 3, 3; SC 442, 324). Neither the debate between medical schools was a problem for Christians, as Clement of Alexandria points out: "the fact that different doctors follow different theories, according to the different schools, but all give themselves to medicine" is not an excuse either "for not calling the doctor just because there is a diversity of medical schools" (*Stromata* VII, 15,89,3; FP 17,510).

Furthermore, despite Early Christianity being configured as a religion of healing[5], consciously or not, and increasing in manifestations of miraculous cures in the dispute for the recognition of the true savior, there is a distinctive feature for which it would be more appropriate to speak of a religion marked by caring for the sick (Ferngren 1992, pp. 8–11, 13). This corresponds to the Christian self-understanding of a fundamental *ethos* as a community organizes itself around love for the neighbor (Theissen 1999, pp. 61–69).

It is this emphasis on an ethical duty of love for the sick in this specific context of healing religion that an exercise of discernment can be found in the search for the desacralized contribution of medical art and its potential to be at the service of the Kingdom of God. It also results in a dialogue with some Stoic schools around the interpretation of Hippocratic naturalism and its application in medical practice. It is here that primitive Christianity inaugurates an intellectual itinerary that starts from the context of the *sanatoria* of the healing religions to reflect on the stoic ideal of *salus*, in which the issue of health is situated as an affinity between the stoic perception of the restoration of nature and the Christian perception of redemption of the same, not restricted to ritual practices, but through the mediation of a *Logos* to be contemplated and that illuminates a way of being. That is the reason why it was chosen here to speak of Christianity as a religion of health that incorporates the stoic ideal of promoting health in its soteriological understanding.

Consequently, Hippocratic naturalism was generally accepted by ancient Christian authors (Temkin 197), who in turn, along with Stoicism, are responsible for the dissemination of the desacralization of disease and cure. From the Hippocratic tradition, Christian authors take up the idea of conformity with nature (Clement of Alexandria, *Pedagogue* II,10,3; SC 108, 164–174) and that medicine must be left to men and not to the gods, and that physicians may be used by God, but are "earth-born creatures" (Arnobius, *Adversus nationes*, I,48; BAC 632, 95).

Another factor favorable to the Christian reception of Hippocratic naturalism is its integral anthropology, as Clement of Alexandria also recognizes: "Hippocrates of Kos says that ascesis is good not only for the body, but also for the soul, and that fearlessness is proper to health, in the face of fatigue and the incessant need for food" (*Stromata*, II,126,4; FP 10, 274). Basil of Caesarea (1968) also agrees "with those who rightly defined health as the balance of natural forces (κατα ουσιν)", and then he adds: "and if the same is said regarding the good state of the soul, it is no less certain" (*Hexaemeron*, IX, 4, 2).

The influence of the spirituality of the *salus* promoted by the Pneumatic School must also be considered in the Christian reception of Hippocrates. However, it is also necessary to recognize a dematerializing tendency there, especially from the influence of Philo of Alexandria's reading of Stoicism as an interlocutor of Alexandrian Judaism.

*5.2. The Spiritualization of the Pneuma in Philo of Alexandria*

Philo of Alexandria develops a new approach to *pneuma* by reading the Hebrew Scriptures in light of Plato, in order to re-elaborate the *pneuma* doctrine and a psychology less tied to Hippocratic anatomy. Starting from the double biblical narrative of the creation of man (Philon 1994, *Genesis* 1.26; and 2.7), Philo distinguishes, in addition to the two stages of the creation of the celestial man and the material man, two dimensions in the latter: the formation of the earthly man consists of an earthly and perishable body and intelligence (*nous*). This *nous*, through the *pneuma* given by God to earthly man, becomes immortal and capable of knowing God and thinking intelligently (Philon 1961, *Legum Allegoriæ*, I, 37).

The *pneuma* in Philo, unlike the Stoics, is not material, but neither is it God, nor a part of the divine, but an expression of the power of God given to the human soul, which gives it the power to think, know and act (Philon 1994, *Quod deterius*, 90). In this way, the Philonian psychology of the *pneuma* comes close to and, at the same time, distinguishes itself from the Stoic anthropology, according to which the soul is a *pneuma*. This *pneuma* given by God to earthly man, which designates the upper part of the human soul, is of an immaterial (but not divine) nature. Anyway, due to its divine origin, *pneuma* appears related to God added to human nature. Thus, Philo demarcates himself from the Stoic materialist immanentism, by identifying the *pneuma* with the upper part of the human soul, which he also calls *nous*, and which is immaterial and immortal, as a mark and image of God. All the more so that alongside the noetic and rational soul (*psiqué nous kai logiké*) similar to Chrysippus (SVF I, 443,2), Philo still conceives of a "sensitive soul" (*psiqué aisthetiké*) of inferior order, material and mortal (close to the *psychic pneuma* of the Stoics). For example, when the human being/soul is in agony and has stopped breathing and the body has already ceased the signs of life, remains for some time in the body, still remnants of the *psychic pneuma*. Therefore, this is not the immaterial and *divine pneuma*, but a *material pneuma* related to vital heat, which the Stoics did not distinguish from *pneuma* (Philon 1961, *Legatio ad Gaium*, 63).

Philo describes the process of knowledge as a pneumatic current that, starting from the "hegemonic part" (SVF 484/4; 826/2; 841; 96 (210; 729; 732; 337), "as from a source", "if it propagates through the various sensory organs", which are the eyes, ears, nostrils, etc. (Philon 1994, *De fuga et inventione*, 182; cf. SVF I, 416; 458).

Under the influence of the Platonic *nous* and some Stoics (Cleantus, Posidonius), and especially the biblical texts, the Jewish Alexandrian philosopher contributed decisively to the *spiritualization of pneuma*, giving it the qualifications (immateriality, proximity to the divine, immortality, indivisibility, intelligibility) which can be founnd in most Christian texts in a Trinitarian pneumatology that develops the personalization of *Ágios Pneuma*. Combining medicine and anthropology, and joining Plato with Aristotle, for example, the Christian apologist Tatian, speaks from a pneumatological anthropology perspective of *pneuma* as the "wings of the soul", while enabling the body to be *capax immortalitatis* through communion with the "Heavenly" *pneuma* (*pneuma tou ouranós*), on his initiative (Tatian, *Discourse to the Greeks*, 20; BAC 602).

Perhaps this passage from Hippocratic naturalism to the spiritualization of *pneuma* was a factor that went underexplored in the 19th century[6], which led to considering the use

of medicine as a repository of metaphors for the care of the soul (Amundsen 1982, p. 331), and less the shared vision of a *salus* spirituality.

Furthermore, it is worth saying that despite this movement, albeit moderate, of *spiritualization of pneuma*, this stance of Philo is closer to Christianity that distances itself from the pneumocentric anthropology of the Gnostics and the Platonic dualism, since Plato often defines the soul with the noun *pneuma*, in the sense of "a diffuse spirit", "prisoner" of the body (*Phaedro* 246a d). Gnostic pneumocentrism, unlike the Hippocratic pneumatic primacy, places the pneuma above the body, while Christian authors, heirs of the Pauline trichotomy (1 Thes 5,23), conceived of the human being as a unit composed of body-soul-spirit, as Irenaeus of Lyon would formulate the *pneuma* being what gives unity to the whole: "The body receives from the soul the breath, life, growth and cohesion" (*Adversus Haereses* II,33,4). Given such an understanding of unity, and far from the dualism that puts the soul and body in opposition, Irenaeus of Lyon can affirm that "like water poured into a vessel, it [soul] takes the form of the body" (*Adversus Haereses* II,31, 1; SC 294, 326), emphasizing the deep connection of both.

### 5.3. Christianity as a Religion of Health

Despite being a moderate movement, the spiritualization of *pneuma* in Philo opened doors for its platonization through the radicalization of the Pauline duality and clouded other theoretical choices for the operationalization of the concept of spirit in a way that emphasized the notion of health as vital integrity of body and soul. Such options are not non-existent in the theological Tradition, but they are less contemplated given the excessive platonization of the West.

Clues to identify such theoretical options can be found in Clement of Alexandria and later authors who tend to see the human being as an extension of the *Cosmos*. There is a mysterious relationship between human beings and the universe, which has to do with the common presence of *pneuma*. The Alexandrian Christian even speaks of a "carnal *pneuma* that moves quickly and is throughout [the human being] through the senses and the rest of the body, which receives the vital force in which the faculty of nourishment, growth and of the whole movement in general" (*Stromata*, VI, 135,3; FP 17, 274). The *pneuma* generates a human-world unity, between *micro cosmos* and the *macro cosmos*, with the human being the image of the world, with the respective correspondence of the physical (health) and moral (*salus civitatis*) fields, and thus seeing the "world as one common home" (Seneca, *De beneficiis*. VII,1,7). The idea is taken up by Christian authors who, like Tertullian, recognize that "the whole world is the common home of all" (*De pudicitia*, 7; CC 2, 1292); or as Minucius Felix (1964) says, "For God, this whole world is one home" (*Octavius*, 38,1). For this reason Minucius also recognizes that, since everything is connected, the human being must strive to pay attention to the whole, "knowing himself and looking around him", "because things are adjusted, connected and linked with in such a way that you know nothing of the [condition] of mankind, unless you have diligently searched the condition of the divinity, and are not able to perform well what suits the citizen, unless you have known this common city" fruit of "divine reason" (*Octavius*, 17,2). Thus, occurs the passage from a "pneumopolitism" to a "cosmopolitanism" and vice versa, which is based on this connaturality between *pneuma-spiritus*, present in the human (*imago dei*) and in the world (*imago mundi*) in connection with the divine Spiritus, the *anima mundi*. The world is the place of communion between human and divine and the *human pneuma* is the "guide of the soul" through which the *divine Pneuma* leads, as the author of the *Apostolic Constitutions* claims (Apostolic Constitutions 2001, VII,34,6; 39,2; VIII,12,16).

In other words, the notion of spirit indicates that the human is able to understand the *Logos* of the world, that is, he is able to grasp the meaning of all things that are deeply connected by the divine Spirit. And it is this spirit that will direct the body (*sarx*) and the soul (*psyche*) by tuning in to the Spirit that illuminates the choices to be made towards the stoica *salus*, which for Christianity is an intrinsic part of soteriology. If Hippocratic medicine is mainly based on human sympathy or affinity with nature (CH, *Airs, waters,*

*places* II, 39–88; LBL 147), the action of *Christus medicus* is especially characterized by this ability to "suffer with" (*sympatheia*) (Basil of Caesarea 1957, *Letter* 31), which reinforces the effectiveness of its therapeutic action, an idea that was not alien to ancient Greek therapy, but without the same depth. This sympathetic closeness and compassion asked of the doctor helps to explain the motives and effects of the incarnation of the Divine Word: "[Christ] became man for us, so that, taking part in our sufferings, he might heal us]" (Justin, *Apology* II,13,4; SC 507, 364). The incarnation is seen as the "Physician's visit" (Augustinus, *Sermo* 360 B, 17). Thus, if the Stoic *Logos* unveils the meaning of life in order to take care of it, the Christian *Logos* unveils the meaning of loving life, so that the Christian can take care of it and thus save himself.

This is how the Stoic salutology is assumed by Christian soteriology and the saving action of Christ is presented both in literature and in paleochristian iconography, in the image of the "spiritual Hippocrates" [*spiritualis Hippocrates*], a designation used by Jerome (*Contra Ioannem Hierosolymitanum*, 38; BAC 685, 514) to talk about Christ. Also, it is indicative of the process in which the rejection of the religious dimension of the Asclepiad tradition, the recapitulation of the old medicinal tradition, and the incorporation of the Christian therapeutic action, which aims at the notion of integral health, also curing diseases of the soul, occur at the same time. The saving action of Jesus Christ becomes also the "healthy action", witnessed by "an innumerable multitude of people who recognize Jesus as the true 'Asclepius'" (Origen, *Against Celsus*, III, 24–25; SC 136, 56–58); overcoming this *deus medicus* who "instead of healing the soul, leads it to perdition" (Jerome, *Vita Hilarionis* 12,3; SC 508, 246).

Furthermore, one of the distinguishing features of the salutary action of Christ and of Christian pastors will be the gratuity: they heal without thinking about profit (cf. Mt 10:8). The *Salvator* Cristo is "the physician who abandons no one" (Augustinus, *Enarrationes in Psalmos* 58/II, 11) and freely helps all (Arnobius, *Adversus nationes*, I, 49; BAC 632, 96). As Augustine reminds us, "when a doctor approaches a very poor patient and finds him in a desperate situation, in this case he does not ask for a reward, but makes the art count" (Augustinus, *Sermo* 175,8,9). *Christus*, the true *medicus*, saves "not only those who frequent the sanctuaries of Asclepius or dwell near, but all those who dwell far away, that is, everywhere and freely" (Arnobius, *Adversus nationes*, I, 49; BAC 632, 96). Although there is a tendency towards a certain emphasis of Jesus' medicine on the illnesses of the soul (Dumeige 1972, pp. 127–29), a more comprehensive health awareness can be identified with the construction of the *Basiliad*, a complex built by Basil of Caesarea that included a monastery, a hospital, and an inn was a true example of the Christian perspective of philanthropy to the poor and the sick. The builindig had still included a leper house, disease considered contagious and the reason for social exclusion of the leprosy (*Letter* 94; cf. Müller 2018, p. 375).

From this compatibility between naturalism and *salus* spirituality, some ancient Christian authors establish an association between healing, the restoration and redemption of the nature, by the healing words and activities of Jesus and connecting the examples of disease and healing in the Hebrew Scriptures. For one hand, this connection elaborated an edifying Christian rhetoric that operated a true expansion of pagan medicine, transforming it into an integral spiritual therapy, a context in which Asclepius Soter was replaced by *Christós Iathrós* or *Christus medicus*. On the other hand, this medical analogy of the *Christus medicus* has a semantic density referring to the diversity of social practices that are related to the medical metaphors used in the different discursive practices of soteriological Christian rhetoric, due to the different understanding of what spirituality really is, and their respective ascetic practices.

Platonic-oriented asceticism favored an excessive intellectualization, privileging the search for the Delphic principle of the *gnothi seautón* or know yourself. Research in the 20th century on Stoicism revealed another type of asceticism that was characterized by the *epimeleia heautóu*, which the Latins translated as *cura sui* to refer to the concept of selfcare. The etymology of *epiméleia* evokes the forms of activities that lead to taking care of oneself

for a growth in the molds of gymnastic exercises, being spiritual as the *melétai* is related to the practices of exercising thought through meditation, silence, reading and writing, but also physical ones such as *gymnázein* that concerns the exercise of the body, including the care of food and rest that aimed at an aestheticization of life through ethical principles of feeling, acting and thinking (Foucault 2010, p. 77). Such spiritual exercises, due to their dynamics of aestheticizing life and the world, implied exactly the opposite movement of the ascetics of Platonic orientations, such as the contemplation of nature in its beauty and singularity (Hadot 2014, p. 311).

Thus, if for certain Christian realms, medical discourse has only a metaphorical function, on other realms the same figures are a concrete expression of a spirituality engaged in the care of the health of the body and soul, and caring for others, especially those most in need of health. There are many elements of the ancient Greco-Roman medical science that Christianity recovered, once sacred medicine was rejected, under the name of Asclepius and other deities, thus understanding itself as a religion of health:

- 　The Hippocratic principle of going beyond symptoms to the causes of suffering (*Epidemics* 6:3) will often be used by Church pastors and spiritual therapists. Maximus Confessor advises "to look for the causes that produce the disease, to find the remedy" (*Chapters on charity* II, 42; Patrologia Graeca (1857–1866) [PG] 90,1000; cf. Augustinus, *In Evangelium Ioannis tractatus* 25,16).
- 　The Hippocratic principle that advises examining the disease from the causes and the whole (climate, quality of air and places, behavior, habits, age) is taken up by Christian authors (Lactantius, *De opificio Dei*, 4.16; SC 213, 124–130; John Damascene, *Oratio in imagines* II,7; PG, 94, 1288–1289).
- 　The care and defense of life as a doctor's duty (*amicus medicus*) who, therefore, sometimes cannot comply with the patient's will to fight the disease (Augustinus, *Sermones*, 9.4; 40.6; 49.6; Clement of Alexandria, *Quis dives salvetur*, XX; SC 537, 150–152)
- 　The analogy of the diagnosis that distinguishes the good doctor, as a "mirror" of the real situation of man, as the first step towards the cure of the disease (Clement of Alexandria, *Pedagogue*, I,9,88; SC 70, 264).
- 　The doctrine of the four humors and the four elements and their role in psychosomatic pathologies (Gregory of Nyssa, *De opificio Hominis*, 14; 17; SC 6, 146; Augustine (Augutinus 1965, *Letter* 205, 3–4; *De Genesi ad litteram*, III,4,6; Nemesius of Emesa, *On Human Nature*, IV-V; Isidore of Seville, *Etymologiae*, IV,5).
- 　The patristic authors take up yet more advice from Hippocratic therapy, applying it now to spiritual ascesis. Thus, the Hippocratic advice of "moderation in food and physical exercise" (*Epidemics*, VI,4,18) is adopted by Clement of Alexandria, recognizing that, "according to the doctor of Delos, the cause of the disease is the excess of food" applying it to the Christian life (*Pedagogue*, I,2,3; SC 70,112; *Stromata*, II,126,4; FP 10, 274) and the same happens with the baths (*Pedagogue*, I,9–10; SC 70, 124–128; Origen, *Homiliae in Psalmos* 37; SC 258).

It is precisely the fact that medicine is totally desacralized that makes it possible to see in it an action of divine providence and a typos or image of the treatment of the soul. Athanasius of Alexandria recalls how distinguished the "physician and savior" Jesus Christ is, who "not only healed the wounds, but also formed nature itself and restored the body to its integrity" (Athanasius, *De incarnatione Verbi*, 49,1–2; SC 199; 444). In this way, the *secularity* of the medical art as "rational medicine" (Weissenrieder and Etzelmüller 269, 275) and the *bodily integrity* are compatible with the spirituality of health that is part of the conception of Christian *salus*. It is within these conditions that Basil of Caesarea presents the medical art as a "gift of God" (*Regra extensa*, 55; PG 31, 1044–1045).

This "conversion" of Hippocratic therapy into spiritual therapy also took place with the incorporation of an integral view of health, not being, therefore, a mere metaphorical use, but also the incorporation of a salutology in the soteriological view. Salvation is implied in integral health in history, and so there was also a conversion from Christianity that operated under the influence of the same Hippocratic tradition and the multiple uses it had in the

context of Greco-Roman culture. Potamius' work, however, shows us more precisely that the Christian reception of Hippocratic art was not restricted to spiritual or metaphorical hermeneutics under the inspiration of the pneumatic school, but also dialogued with the empirical medical school. This is why Christian priests came to presented themselves as therapists of body and soul (Saba 2012, pp. 404–29).

## 6. The Hippocrates' Reception by Potamius of Lisbon

The success of the Greek-Roman "deus clinicus" also reached the peninsular territory, in the city of Olisipo, now Lisbon. Archeology has found three eloquent testimonies of the worship of this divinity, in such a way that the hypothesis of the presence of a temple to the thaumaturgist god is not rejected, especially due to the vestiges of a theater, typical of the Asclepiades Temples. In this context arises the reflection of Potamius of Lisbon, the first known bishop of Lisbon, who died in the year 360 AD, at around 60 years of age. The initial involvement in the Aryan controversy ended up marginalizing him among the great authors of Orthodoxy, from which he was later rehabilitated. However, the rediscovery of his works brings important contributions, especially in dialogue with the Hippocratic tradition.

### 6.1. Anatomy of the Trinity and Rhetoric of the Body: From Homo Imago to Imago Dei

In the 3rd-4th centuries AD it is not uncommon to find medical bishops and priests. By way of example, one can cite Eusebius of Rome (309–310), physician and physician's son, and bishop of Ancyra under Constantine; Constant, Bishop Theodotus of Laodicea; Eusebius, Bishop of Laodicea (257) (Eusebius, *Historia Ecclesiastica*, VII,32,11; SC 41,225), not to mention the fame of St. Gregory, the bishop "thaumaturge", a witness of "great hope of health" (Gregory of Nyssa, *Life of Gregory Thaumaturge,* 78; 98–99; SC 573; 196; 222–226).

In the case of Potamius, everything indicates that he had completed the common medicinal education with some specialized readings. He corresponded with Athanasius of Alexandria, who also frequently uses medical language to refer to the philanthropic and salvific action of Christ (*On the Incarnation of the Word*, 18; 44; SC 199, 330–332; 424–428), and it is very likely that he was along with the Alexandrian medical school which enjoyed a good reputation in the 4th century (Scarborough 1969, pp. 141–42), based in Alexandria. It is significant how the appreciation for Potamian anatomy brings it closer to the human anatomy of Alexandrians or empiricists. (Nutton 2018, pp. 144–56).

It is in this cultural and mental atmosphere that his work *De Lazaro* is inserted, where Christ appears as the new doctor and "true friend of men": in the path of an already long tradition of using knowledge in the field of medical science, present in the authors of the first five centuries which is no different from what was common knowledge in the Greco-Roman world in which they lived (Ferngren 2009, p. 13); a Christian tradition in which medical knowledge and scientific vocabulary have already taken on a strong cultural connotation; in a Christian culture in which nosology acquires, however, a new anthropological and sociocultural charge: "disease", "health", and "contagion" have a social charge that goes far beyond the scientific one. Heresy, for example, is seen as a contagion that disturbs the social order (Potamius, *Ad Athanasium* 1)[7].

It is from the figure of the physician Christ (*amicus medicus*) that in Potamius' work the presence of the Hippocratic tradition becomes evident again. With this we do not mean to suggest a direct connection/reading, since Potamius' medical baggage supposes the later hermeneutic tradition suffered by the *Corpus Hippocraticum*. But this does not dispute the fact that Potamius is inscribed within a Christian tradition receptive to the contribution of medical science to Christian theology as *science* that better helps to understand the meaning of redemption from the analysis of the restoration of nature from the empirical medical practices, not only as imagery for theological notions.

As a rule, most patristic authors use medical terminology in merely metaphorical terms for thematization and understanding of religious ideas. However, in the case of Potamius, medical art in general, and anatomy in particular, serve as a validation of his

theological argument. In three of the four brief writings written between 340 and 360, the first known bishop of Lisbon makes use of anatomical and medical notions in support of his doctrinal arguments. It is significant that he does so in his preaching (Sermons: *De Lazaro* and *De martyrio Isaiae*), as in a treatise with markedly theological content (*De substantia*).

After having presented the Son-Christ as the Father's agent in the role of a doctor: "who raises Lazarus, who takes away the heat of fever, who unties the knots of the paralytic, who stops the blood in the veins . . . who makes the troughs move, who it makes the dumb speak, it gives ear to the deaf, it gives sight to the blind" (*Substantia*, 35), having addressed an enthusiastic exhortation to the *fraternitas* as "people united to the Lord God friend" (*Substantia* 40), in the last section, Potamius describes the unity of the substance of the Trinity using images taken from the human body, taking as its starting point the sentence in the book of Genesis (42 69): "Let us make [the human] in our image and likeness" (Gn 1,26; (*Substantia*, 41. 43. 45). For Potamius, the human being is the image of God not only as a spirit (*pneuma*) or consciousness (*noûs*), but in the bodily constitution itself. He then proceeds to demonstrate how the knowledge of God "is apprehended in the face of the human being":

> "The knowledge of the Father and the Son is apprehended in the face of man, the Father and the Son being just as the shape of their face is imprinted on the human archetype made of clay, with which we are modeled, so that man could admire God from the beginning of the human being". (*Substantia* 41)[8]

Potamius  (1999) shares the idea, quite common in the 4th century, that the nature of the human being, and knowledge of it, from a medical point of view, consists of recognizing that the human was created in the image and likeness of God (Hamman 1987, pp. 153–237), but in a different way of the tendency to restrict the *imago Dei* to the soul or the activity of the spirit (*noûs*), in the line of Plotinus (1960, 1 *Ennéades*. VI,7). Potamius, therefore, while disagreeing with the dogmatic Medical School, sees no inconvenience in mixing the traditional Platonic anthropology with the data of medical science from the Empirical School, to see the image of God in the physical constitution of human beings, along the lines of the anthropology of Irenaeus and Tertullian, in which it is the concrete human being, created and modeled from the earthen clay that is *imago Dei*. It is especially in the *vultus hominis* that the *vultus Domini* (*Substantia* 42) is mirrored, for "the human face manifests the image of God and piety enters the human through the image of divinity (*vultus humanus ostenderet . . . imago divinitatis*)", God, "when he modeled the human creature", imprinted his own face on it (*Vultus Domini praefetur in homine*, *Substantia* 42). The Bishop of Lisbon continues:

> "Therefore, when God said: Let us make the human being in our image and likeness, He wanted to see what He is capable of, when in the human He revealed who He Himself was. He said "our" and a single face was formed, to indicate that the human being has the features of the Father and the Son". (*Substantia* 43)[9]

Based on the founding text of the Book of Genesis 1, 26, Potamius sees the image of the one and triune God in the external features of the human body, especially in his face; that, despite the dissonance between the faces, the experience of contemplating the face is the same (*Dissona facies, sed vultus aequalis est*, *Substantia* 43). Establishing a dialogue between biblical anthropology and Hippocratic medical physiology, the Potamian argument is developed based on offering evidence that, in the human make known how God is, from the unit-plurality binomial, that is, the Unit of Substance [*divine*]-distinction of divine Persons in parallel with the unity of the senses, plurality of sensory organs:

| | |
|---|---|
| *Duo sunt oculi—sed unus aspectus est* (44) | The eyes are two, but the vision is one (44) |
| *Sic et naribus . . . duae videtur esse personae—sed in ipso supercilio una probatur esse concretio* (50) | Also the nostrils . . . look like two individualities—but because of their own external shape it proves to be one (50) |
| *Nam et oculis duae gemmas habere dicimus—sed unus est visus* (51) | In the eyes there are two pearls, but only one vision (51) |
| *Genae ipsae . . . duae dextera laevaque videtur in facie—sed ne aliquid separentur . . . ut tota unius substantiae . . . nerretur* (57) | The cheekbones themselves . . . appear on the face as being two—but so that they can in no way be separated . . . so that the whole reason of a single substance . . . makes itself known (57) |
| *Duae aures sunt—uno sibimet traducefibulatae* (59) | There are two ears joined together by a single communication (59) |
| *Separati sunt digiti—sed protinus conexi* (67) | Fingers are apart—but closely joined (67) |

The detailed analysis of human physiology, from the face to the hands, directly linked to *thesaurus cordis* (*Substantia* 43–67), allows the author to move from *the imago hominis* to the *imago Dei*, through clinical dissection, in profane terms, of the "sacred face" (*sacer vultus, Substantia* 44), which manifests itself in the unity between all parts. The concept of connections (*conexio-sympatheia*), so important in the Hippocratic pathophysiological and anatomopathological conception for the organic unity of the human being maintained by the connections between different parts of the body (*Places in Man*, 8, 95; LCL 482), becomes a key notion for Potamius (*Substantia* 2; 3; 18; 38; 44; 50; 61; 63; 65; 66; 67).

The action takes place in the diversity of organ functions, and thus describes the physiology of the parts emphasizing the deep connection. This time, the unity of vision takes place in the diversity of the eyes (44–47; 51–52); the smell through the nostrils (47–48); the word through the mouth, tongue and lips (53–55); hearing through different ears (58–62). Then, there is a description of the upper limbs, which also act as a unit of touch (arms and hands, 64–69), following the dynamics of traditional medical science (CH, *Places in Man*, 8, 89–136; LCL 482). All these human organs and faculties coexist and work in coordination, because there is a relationship that unites them (68–69), thanks to the intention of the Creator who made the human being in his image. Therefore, all unity can be seen in the face: "A face is quickly identified by the hearing of its ears, the sight of its eyes, the breathing of its nostrils and the truth of its tongue" (*Substantia* 59). Based on the biblical affirmation of the book of Genesis, Potamius develops a trinitarian anthropology, as the creator "imprinted his image on the human face" (*Imaginem suam in hominis vultu signavit et dixit: ad imaginem nostram*, cf. *Substantia* 41).

In describing, for example, the anatomy and mechanics of the ear, Potamius comes very close to the anatomical description of the Hippocratic treatise *On Meat* (18). Apparently bearing in mind the Hippocratic precept, according to which "the body is the starting point of medical discourse" (*Places in Man*, I, 1; LCL 482), from the functional unit of the sensory organs, the bishop of Lisbon evidences in human, the *imago Dei*, as unity of the divine Substance. With the purpose of "demonstrating, from the very reason for being of things, that it is clear that there is only one substance, both in the eyes and in the nostrils, in the mouth or in the ears, linked without a doubt by the bond of union and centered on the single forehead of the head" (*Substantia* 61). In describing the physiological organs of the other senses, Potamian physiology serves as an epistemological basis for a theological analogy. Potamius goes from the body to the theological discourse: from anthropology to theology, from man as the image of God in his bodily configuration, to the consubstantial Trinity:

"What one [ear] catches, immediately sends it to the other. And from this same one, which he had transmitted after retouching it, he quickly returns again to the other one. The breath immediately makes them both vibrate: one, what he caught, and the other, what he heard. Everything that one perceives, soon passes to the other. If you praise one, through the other both will know everything you say. One communicates to the other what was heard, the fidelity of one does not

retain what is owed to the other. Both reproduce the only thing anyone has said. With reason the ears always have a single hearing of indivisible solidity, because, through his image, which God incorporated in man, not separating but uniting, by virtue of their likeness the indivisible Trinity with the Son was represented by the Holy Spirit". (*Substantia*, 59–61)[10]

In the biblical anthropology of the *homo imago Dei*, Potamius sees the representation of the Trinity in anatomy itself, and not in the internal and intellectual faculties, as Augustine does. Based on empirical observation, he develops first in the foreground, an anthropo-biological and anatomical plane, and after in the background, the spiritual anthropology, in which there is a somatic connection between "the faculty of the senses gathered in the face, because it is the image of God" (*Substantia* 62), and so the nostrils, mouth and ears speak of the Trinity. In this anatomy of the Trinity, Potamius follows the usual order of medical anatomy, starting from the head/brain as the center and seat of intelligence and culminating in the heart, seat of the virtues (*Substantia* 63; CH, *Fleshes*, 4; LCL 482).

*6.2. De martyrio Isaiae: Anatomy of Death and Body Spectacle*

Given the empiricist orientation of the Hippocratic reception in Potamius, in the two sermons to be analyzed, the Lisbon author, with parenetic purposes, uses medical science to elaborate an anatomy of death in different perspectives: to exalt martyrdom (of Isaias); and to preach the resurrection (of Lazarus). In either case, the focus is on the body and not the soul, meantime with the soul always present. It is possible to note, first of all, that the two Sermons have an identical structure and rhetorical construction: contrasting the decomposed (Lazarus) or disemboweled (Isaiah) corpse with the martyr's resurrection and victory. In both cases, the space dedicated to the anatomical description of the bodies in question is noteworthy, in order to highlight the triumph (of the martyr, and of Christ) over death and *miseria corporis* in both (*Lazarus* 11). In both cases, it is a *sui generis* thematization of death, from the perspective of victory (of the martyr) or of the resurrection (of Lazarus); two modes of triumph over death, in which the body is seen as a "place" of combat.

Both the resurrection of Lazarus and the martyrdom of Isaiah are staged as a spectacle (there should be witnesses to the miracle) and, at the same time, in the manner of an anatomy class. The author seems to know the attractive force of the exposed corpse or the limits/borders of life. Isaiah's martyrdom is described as a *caelestis pralestra*, a great miracle (*miraculm*), a shocking *spectaculum* to behold, and yet the preacher asks them to "pay attention with a solicitous eye" to the "old science that deals with sufferings." The aim is to inculcate an appreciation for the "nobility and remarkable vigor" (*martyrii nobilitas . . . praeclara . . . fortis*) of martyrdom (*De martyrio Isaiae* 2).

In the context of the 4th century, one speaks of the bloody *spectaculum* of the martyrs of the first three centuries, given the emphasis on the marvelous and the heroicization of the martyr (Tertullian, *De spectaculis*, 4; CC 1, 231; Cyprian, *Letter* 60, 2, 4; BAC pp. 717, 732; Ferngren 2009, p. 78). However, the dramatization of the Potamian scene is not centered on the martyr's torments, but on the body as such, presenting a new representation of violence, which no longer aims to document the physical violence exerted on the martyr, who engage in a physical *agon* as a fighter for faith (*miles Christi*), as in the old *Acta* and *Passiones*, but it is now a new source of interest for listeners. In the case of *De martyrio Isaiae*, the martyr's blood ceases to be a witness of *imitatio Christi* (and, as such, "seed of Christians"), to be "admired" (*miraretur*) by the audience (*De martyrio Isaiae*, 4).

Martyrdom as a spectacle, which scandalized the moderate stoic Marcus Aurelius (1916), who criticizes its "theatricality" (*tragôdios*) (*Book* 11.3), is now used as a rhetorical device to arrest listeners' curiosity. Throughout the speech and as the martyr's body is divided in half by the executioner's saw, starting with the head (3), reaching the heart (4), with the hyper-realistic description of the "labyrinth of the veins" and "of the whirlwind of blood" (4), the verb *miraretur* (4; 5) emphasizes the wonder and reason for admiration of human anatomy, just as the author of the Hippocratic treatise manifests analogous admiration when "observing the heart" as "the work of a skilled craftsman" (CH, On the

heart, 8, 181). It is particularly unheard of how the author explicitly uses this anatomy of death (*veteris disciplinae*) to celebrate Isaiah's *passio* as a spectacle worth seeing (*de passionibus celebritatem advertite*, *De martyrio Isaiae* 2).

The term *advertite* appeal clearly announces the author's intention to make his audience participate in what is about to happen: a kind of anatomy class. This begins with the "dissection" of the head, that is, in the words of Potamius himself, "at the center where the sensitive faculties meet" (*De martyrio Isaiae* 3).

### 6.3. De Lazaro: From the Anatomy of Death to the Anatomy of Life

The figure of *Christus medicus* appears especially in the Potamian narration of the episode of the resurrection of Lazarus, who, like Galen, supported the same idea of Hippocratic philanthropy in which the doctor begins treatment through a relationship of sincere friendship (CH, *Precepts*, 6; LCL 147; Ferngren 2009, pp. 88–94).

Furthermore, in the narrative of the resurrection of Lazarus, Potamius follows a description analogous to that in *De substantia*, of the senses and functions of the human body: sight, hearing, voice, and movement of the limbs. In contrast to the anatomy of death described in the first part (2–8), the anatomy of life is presented in the second part. Not long before Potamius, Lactantius had already given a description of the sensory organs, underlining their double unity (*De opificio Dei*, 10.9; SC 213, 160. 162). Lactantius' description seems, however, more attentive to the data of medical science, including when he describes the respiratory system (*De opifício Dei*, 11, 3.5; SC 213, 170). Potamius' anatomical detail is less faithful to science and more geared towards rhetorical effect. Hence its "hallucinating anatomy" and "tenebrist" that appeals to all the senses, emphasizing sight and hearing in *De Isaiae*, and smell and vision in *De Lazaro*, which highlights the "medical pedagogy" of Potamius standardized as an anatomy of life, as the final part of *De substantia* illustrates. Potamius is a great representative of the symbiosis that has been produced between Hippocratism and Early Christianity (León 1998, pp. 519–21).

However, the Lusitanian author does not limit himself to making a metaphorical and superficial use of medicine for illustrative purposes, but his sermons show an approach to the medical pedagogy of his time, mainly influenced by Galean, in making anatomy a rhetorical practice (Von Staden 1995). Galean thus inscribes himself in the rhetoric of the so-called "Second Sophistry" (Von Staden 1997), in order to rescue the heuristic and experimental tool of the empiricist school, which had lost ground to dogmatist tendencies. In this way, anatomical exposure also gains a political connotation of influencing medical culture in particular, but also the culture in general of correcting abuses regarding the conception of health and its natural causes. Thus, the use of rhetoric in Potamius indicates both the adoption of a specific medical pedagogy and an empiricist epistemological inscription to think about theological questions. Potamius' originality resides in the rhetorical power (*vis*) he attributes to medical art for the elaboration of his Christian anthropology and dogmatics based on an integrated view of the body and soul.

If most patristic authors use the Hippocratic heritage to talk about the health of the soul (Larchet 2000, pp. 36–46), Potamius starts from the body and remains in the field of biological anthropology, exalting the representative dimension of the body, not just as *imago Dei*, but also in its radical fragility (*forma et fabula*, *Lazaro* 2). Henceforth, the wonder of man does not lie in being *imago mundi* (microcosm), or, in the Hippocratic version, in participating in the four elements (Lazaro 8), but in being *capax* of resurrection (*ut viderent na ressurgere possit Lazarus*, *Lazaro* 13), because of the *sympatheia* of Christ.

### 6.4. Corporis Fabrica: Human Nature

The decomposition of the nature of the *Corpus* (*Lazaro* 7) results from *Divortio* or separation (*distantia rerum*, *Lazaro* 7) from the constituent components of human life. It is, however, the only place where Potamius approaches the already consecrated tripartite anthropology. His fascination with the body is such that the other dimensions of man seem forgotten. The term *Corpus* is one of the most recurrent (18 times). In *De Lazaro*, he uses the

technical expression "*corporis fabrica*" twice, which refers to the anatomical organization of the body (6; 8). Before exposing the miracle (*spectaculum*) of the resurrection, he focuses on the analysis of the human body/corpse, and on the anatomy of death (2).

In the Hippocratic treatise *On ancient medicine*, the author begins by disagreeing with the ancient "wise doctors" who maintain that it is "impossible to know medicine without knowing what man is" (20; 161; LCL 147) to argue that "only from medicine is it possible to know something for sure about the *physis* [of the human being]" (8, 29). This claim seems to be implicit in Potamius' form of arguing that confesses his empiricism in terms of the *fides* comes from the "admiration of the eyes" (*tota fides est hominis quidquid oculus admiratur*, *Substantia* 47).

Faithful to this assumption, it also seems to start from the *physis* of the body, as a particular expression of the *universal physis*. He too knows that illness and death consist in altering the order and dynamics of the body's nature and humors. Potamius even recovers the Hippocratic analogy of flowers that change color and odor when they die (*Lazaro* 41; CH, *On the humors*, 1; LCL 150). While remaining faithful to Hippocratic anthropology, the pastor of Olisipo sometimes associates the *spiritus* with the Platonic image of the *anima*, which compares it to a chariot drawn by two horses guided by a coachman. If the two horses represent the two lower parts of the soul (concupiscible and irascible), the charioteer represents the rational soul that must order and guide the other parts of the soul (*Republic*, 4, 437b–441a). In this sense, the author seems to look at the debate between medical schools and how both can be compatible with the typically Christian tripartite view of the human being: *corpus-anima-spiritus*. The use of the soul that "cultivates the body's flowerpot/garden" while keeping the elements of nature together seems to be in tune with this tripartite anthropology:

> "Therefore, the bodily edifice is composed of earth, water, cold and heat (four parts that, always fighting each other, ruin the body by fighting their demands: heat does not like cold and cold is burdened by heat—the contrary things give in to their opposites—the earth is damaged by excess water and the water gets dirty with the earth). These four-shaped elements forming a single reality with a four-part mass, after the coachman, who had determined the places of these four parts, had been removed by divorce from death, so that none of them could develop too much mobility, and who guided this bodily edifice with domineering whip, divided its boundaries into the whole of a harmonious composition; these four elements, I say, as soon as the soul departs, they are mixed in the amalgamation of the uninhabited body, because the agent of union has been cast out". (*Lazaro* 7)[11]

The *anima* here performs the unifying function of the four elements of the *fabrica corporis* in a harmonious composition (*in unum concordante*) that keeps each of them in its place. Once the soul moves away (*recent anima*), a dissociation of these elements *per divortio mortis* takes place, and the "widower body" (*corpus viduatum*) decomposes. Earlier Potamius had attributed the same results to the withdrawal of the *spiritus* which seems to coincide with the superior soul (*Spiritus sapientiae*). A similar role is attributed to the soul and *pneuma*: *recedentis spiritus* (6) and *recedent anima* "which cultivates the garden of the body, all matter decays through putrefaction" (7). When the *anima* and/or *pneuma* withdraw, the unity and balance between the bodily elements is broken, resulting in corruption/decomposition (6). The "*distantia rerum*" causes the decay of the *membrorum natura* (7). And the very air breathed by the lungs becomes fetid (*foetentia pulmonis spiramenta*, *Lazaro* 4).

Death is a fact, or even an inevitability, for believers as well, as Clement of Alexandria notes: "for all men the end of life is death" (*Stromata* VI, 22,3; FP 17, 96). In this context, the Christian discourse on the resurrection starts at the same time from the paradox of the real misery and dignity of the body. The death and consequent decomposition of the body is something that the "earthly" man, according to the Genesis account, cannot escape (2). As Lactantius, whom Potamius seems to have read, recalls, "God knew that the being he had created naturally tended toward death" (*De opficio Dei*, 4,2; SC 213, 124). Death is this "separation of the parts" (*dissolutio*) that Potamius describes in great detail. Thus, we are

witnessing a kind of theatricalization of death as the first act of life. The "spectacle" of death that causes awe and curiosity begins with the deterioration of the flesh (*Lazaro* 3), but this does not break the *sympatheia*, since the dissolving elements of the body, the four humors and the four elements of the cosmos of which the being human is part, return to earth after death. Death, said like this, can be understood as the moment one enters life, and the spectacle of the resurrection unveils such a deeper meaning.

Thus, after the *spectacle of death* painted in hyper realistic colors, the *spectacle of the miracle* of life follows, in the staging of the resurrection that has Lazarus and Jesus as the main actors: "And I ask you: why did the people go there in such a hurry? What crowd of spectators was that? What kind of spectacle could give such a great miracle?" (*Lazaro* 2)[12].

The spectacle of the miracle now takes place in the spiritual plane that is connected to the biological plane of the return of the elements to the world, and that, therefore, all mortal flesh will once again become the clay of the image of God. The gloomy and tenebrous scene of death/sepulcher (3–4) now gives way to the most luminous *claritas* brought by *Salvator humani generis* (14), which is still associated with the tears of humanity mourning the death of the children of Adam (15). The final part (15–18) offers us the most moving page of the Potamian writings, referring to the sympathy of the spiritual plane that was expected of the *medicus* in the ancient therapeutics. The Christian hope, in Potamian rhetoric, that God does not abandon the human being at the critical moment of death resides, therefore, in the fact that "Because of the death of man Christ became mortal" and thus neither did the Father nor does the Son abandon the deceased: "But neither the Father nor the Son ever despised thee" (16). As a "medical friend", the Lord shows himself close, taking part in the human suffering: "Mary was crying and Sister Martha was sorry for her brother's death. In the face of their tears . . . the Savior, piously moved, responded with his weeping to their weeping". Additionally, in Potamius, God the Father, is also involved in the "weeping" of the Son: "also his Father, in Heaven, was moved by the tears of his Son, our Savior" (*paternitas fletebatur*, 17). The climax of the *spectaculum* coincides with the arrival and intervention of the Savior, who now occupies the scene. The people's increasingly curious eyes are now fixed on the *secretarium horroris*, that is, the tomb where Lazarus lies: "The curious [*curiosi*] eyes of the moaning people turn immediately to the tomb" (18). And behold, the voice of the Savior is heard (*Ecce vox Domini Salvatoris*).

Thus, following the same logic of the double plane, the first part of the discourse in *De Lazaro* begins with the affirmation of biological death: "*Lazarus mortuus est*", and the beginning of the second part of this death-resurrection diptych corresponds with the expression: "The Lord . . . said that Lazarus had fallen asleep" (*Dominus . . . dormisse Lazarum dixit*, *Lazaro* 9). This turning point is due to the intervention of the Lord (*Salvator*, *Lazaro* 10; 11). Death becomes *dormitio* and the darkness of death gives way to the light of life (*Lazaro* 14). And then the spectacle of the narrative turns to the "*caeleste miraculum*" (19), which announces the "*homo redditur*": "death is conquered, man is restored, the chains of hell are broken" (*Mors vincitur, homo redditur, inferorum catenae franguntur*, *Lazaro* 20).

From that moment on, people no longer speak of corpses or bodies, but of *homo*, since only a living body can be called a human being. But it is through the body and its physical functions that man expresses himself as such. That is why *De Lazaro* ends up recovering the anatomy of life:

> After four days, Lazarus' tongue moves again, his hands get ready for work, his eyes turn in their sockets, his steps start to leave footprints again, his hearing returns to his ears and his gaze returns to his relatives. The revitalized sight recognizes family members and your family's voice penetrates your ears. (*Lazaro* 20)[13]

As a result of this spectacle of life/resurrection, there is the restoration of the "outer aspect" (εἶδος) of the body (Gregory of Nissa, *De opificio Hominis*, 27; SC 6, 210–212), a refiguration of the *hominis* figure, thanks to the harmonization of elements whose disorder had caused illness and death, as is characteristic of the presence of *Ágios Pneuma*, who works in the Christological Word (*Logos*), to unite all things and thus give life. The resurrection ends up being presented as a miraculous cure that resulted in the recovery of all bodily

faculties: mouth; ears, eyes . . . And so Christ is presented as the distributor of the *salutaria*, the salutary remedies, not only for the body, but for the disorder.

Faith in the resurrection of the flesh in Potamius recovers the value of the body, placed second from the Pauline writings and its patristic echoes (1 Cor 15:50–53; Irenaeus, *Adversus Haeresis*, V, 6, 1; SC 153, 72–80). From the Potamian point of view, the resurrection is the result of the Christological *symphateia* that is only possible because the savior also took on a body, and life returns with him in its concreteness: "The Lord Savior presents himself to men with a body, because it had clothed itself with a body" (*Epistula ad Athanasium* 7).

A dimension to be explored in Potamius is the potential that the narrative of the Resurrection has in linking the restoration of nature with Christian redemption, from the Stoic *sympatheia* thinking the issue of health also as a social value, and the spirituality of health as a political spirituality of ethical commitment to the sick. Returning to a healthy life coincides with a return to society and corresponds to the notion of integral health in Early Christianity, but it is also connected with the idea that Christianity is a religion for the sick, which unfolds from the fundamental value of loving one's neighbor. The resurrection of Lazarus is at the same time the *individual* restoration of the Lazarus' body, but also his [re]integration to the *social* body, as the Jesus' healing practices in the evangelical tradition suggests a social liberation (Weissenrieder and Etzelmüller 2016, p. 265; cf. *The Illness Construct of Leprosy (lepra) in the Gospel of Luke*, pp. 280–282). The idea of the body is evoked in the *Letter to Diognetus* around 120 AD, written by an anonymous Christian to an educated pagan who wants to know how Christians relate to Greek-Roman culture and society. There the *body* corresponds primarily to the *city*, and "what the soul is in the body [ἐστὶν ἐν σώματι ψυχή], that are Christians in the world", and like "the soul is dispersed through all the members of the body, and Christians are scattered through all the cities of the world", being the Christian in the city [*polis*], a citizen [*politai*] (Ad Diognetus 2001, pp. 5–6).

The Ressurrection narrative applied to the health issue in Potamius based on the christological *sympatheia* category helps to clarify the close link between health and society, represented by the sisters, friends, and the multitude of people who witness the miracle there and receive it, a strong discursive support of the one of the Christianity's greatest contribution to medicine, the "establishment of caring communities" (Retief and Cilliers 2006, p. 275), as a dimension of social engagement inherent to the concept of Christian health, as "healing community" (Weissenrieder and Etzelmüller 266).

## 7. Conclusions

The concept of *pneuma/spiritus* has undergone a great semantic evolution since Hippocratic naturalism, which gives the term the idea of a vital force or the very vitality of life as its founding and guiding principle. The use of *pneuma* in the Stoic conception of cosmological *sympatheia* offers the Pneumatic Medical School the notion that everything is interconnected, thus conferring on it the notion of integral health and a spirituality of self-care as an expression of harmony with the Cosmos, the human being as a micro-cosmos, an *imago mundi*.

With the rereading of the Stoic *pneuma* under the influence of Philo of Alexandria's Middle Platonism, the notion of *pneuma* is added as a theological element that will be appropriated by Christianity and elaborated in a *pneuma* hypostasis, that is, in a divine personification. This is the context in which the Hippocratic therapy also moves towards spiritual therapy, without losing the Hippocratic naturalism that allows for a spirituality of health that at the same time desecrates the causes of illness and cure and values the role of medicine as a gift from God. On the one hand, some Christian variants that will return throughout history, whether to a posture of theodicy or to a position of privilege of the soul over the body, reside in this Platonic-Philonian hypervaluation of the spiritualization of *pneuma*, understood in a fragmented way. On the other hand, there are other ancient Christian authors, who along with the stoicism were the main responsible for the de-divinization of medicine.

However, the anthropological fact of this theological construction is that *pneuma* starts to be understood in the dynamics of a dialogue between divine and human *pneumas,* thus configuring the elements for what will be later named in modern times as the voice of conscience and consciousness, which will be considered the seat of decisions, place par excellence of freedom that is not affirmed for being *free from* conditioning, but for being *free to* decide on the human condition, a place where *Logos* is greater than logic, and constituting a true sense that guides all existence. Here the religious question should not be confused with the therapeutic one, but a possibility of a positive reading would reside in the fact that an authentic religious experience, while seeing in it a deep meaning of life, has therapeutic effects. The same goes for the therapy that, being positive, expands the possibilities of a healthy religious experience.

Finally, in Potamius, the semantics of the spirit as a theological category is unusually revisited by the recovery of the Hippocratic tradition through the Empirical School, thus emphasizing the relationship of integrality between body, psyche, and consciousness as a kind of antidote to the tendency towards *spiritualization of the pneuma*, to accentuate its aspect of integrating, vitalizing and unifying pneuma dynamism. In the Potamian conception, pneumaticity is what gives vitality to the health of the body; in other words, the development of a spirituality as a search for meaning (*Logos*) of the health allows the conception of Christian soteriology applied in an integral way on the selfcare practices and the care for others. Such integrality presupposes the philanthropy, at the same time Hippocratic and Christian, of collective health care. The resurrection narrative, as a pneumatological narrative, is deeply empathetic in providing health to the social body, as it is not limited to the act of reviving the body but ensures the reintegration of the person as a whole, which includes society and the city common. Both in Stoicism and in Christianity, the notion of *pneuma* provokes the search for deep correlations between the innermost and the most universal, and more precisely in Christianity it was also involved in seeing the connections between the first and the last in society, connecting the common health and Common Home through a political spirituality of selfcare and care for the sick.

**Author Contributions:** The Authors have contributed equally to this work and share first authorship. Currently, both are developing their research as members of the Working Group on Epistemology of the Common House, within the scope of the project «Common home and new ways of living interculturally: Public theology and ecology of culture in pandemic times» linked to CITER. All authors have read and agreed to the published version of the manuscript.

**Funding:** This research received external funding from FCT—Fundação para a Ciência e a Tecnologia, Portugal Government, COLIVING20 FCT/CITER.

**Data Availability Statement:** Not applicable.

**Acknowledgments:** Our acknowledgement to FCT—Fundação para a Ciência e a Tecnologia, Portugal Government and CITER UCP, for supporting the research project that resulted in this publication.

**Conflicts of Interest:** The authors declare that the research was conducted in the absence of any commercial or financial relationships that could be construed as a potential conflict of interest.

## Notes

[1] For the Hippocrates' works presents in the historical collection usually called Corpus Hippocraticum [CH], it will be used here the Hippocrates (1923–1931) LCL: The Loeb Classical Library cf. Vol. I-IV: LCL 147–150; V: LCL 472; VIII: LCL 482.

[2] *Aesculapius* is the latin version to the Greek name *Asklépios* (Ἀσκληπιός).

[3] *Das Evangelium vom Heiland und von der Heilung* was the expression used by Adolf von Harnack in his famous pioneering work called *Medicinisches aus der ältesten Kirchengeschichte* of 1892, and later taken it up again in his work *Die Mission und Ausbreitung des Christentums in den ersten drei Jahrhunderten* de 1902. However, the French and English translations translated by *The Gospel of the Savior and of Salvation*, exploring the more soteriological semantics of the German radical *Heilung* that can be used both to save and to heal. Ferngren (1992) also explores the healing dimension, as suggested by the context of the German theologian's book.

[4] References from ancient Christian authors will be taken from the collections: *Biblioteca de Autores Cristianos* (1954); *Corpus Christianorum—Nova Patrum Series Latina* (1953); *Fuentes Patrísticas* (1992) Patrologiae Cursus Completus, Series Graeca or or

more commonly known as *Patrologia Graeca* (PG); *Sources Chrétiennes* (1941) and Nuova Biblioteca Agostiniana (Augutinus 1965). Due to the large number of published volumes, we chose to refer to the collection by the initials indicated here and the volume number in question, as well as indicating only the year the collection began in the bibliographic references.

5    for a critical analysis of this healing religion conception of Harnack cf. Weissenrieder and Etzelmüller, *Illness and Healing in Christian Traditions*, pp. 263–305; Ferngren, *Christianity as a Religion of Healing*, pp. 64–85.

6    Although Harnack, for example, recognizes that the soul for the Stoics has corporeality (Von Harnack 1892, p. 43), he presents them fundamentally as paying attention to the "health and diseases of the soul" (Von Harnack 1906, p. 91).

7    For the potamian latin textes we use the editing of M. Conti, *Potamii episcopi Olisponensis opera omnia*, CC 69A.

8    Free translation cf. "Patris et Filii cognitio in hominis vultu digeritur et qualis essent Pater et Filius, talem in archiotipam humanam de limo, quo fingimur, caracter sui vultus expressit, ut homo Deum ex homine miraretur".

9    Free translation cf. "Ergo cum dixit Deus: Faciamus hominem ad imaginem et similitudinem nostram, voluit videri quid possit, cum in homine qualis esset ostendit. Nostram dixit et unam faciem fabricatus est, ut Patris et Filii hominis lineamenta signaret".

10    Free translation cf. "Una, quod accipit, in alteram mox refundit. Ex ipsa illa, quae fuerat quod coloraverat, mox recurrit in alteram. Ambas mox vibrat spiritus, unam, quod recipit, aliam, quod audivit. Quicquid alia conceperit, mox et in alteram transit. Si unam laudas, per aliam quicquid dixeris ambae cognoscunt. Una uni infundit auditum, fides alterius quod debetur alteri non fraudat. Unum ambae, quod quisquis dixerit, repraesentant. Merito indiscissae soliditatis ipsis auribus unus semper auditus est, quia per imaginem suam, quam Deus in homine non separando, sed unitando collegit, suae similitudinis cum Filio per Spiritum Sanctum indiscissam reddidit Trinitatem".

11    Free translation cf. "Terra igitur, humore, frigore et calore composita corporis fabrica (quae quattuor partes semper sibimet repugnantes corpus criminum in procella subvertunt: calor frigus non amat et frigus calore torquetur—contraria contrariis mancipantur-, terra nimio humore vitiatur et humor de terra sordescit), his quadriformibus elementis in unum quadripartita mole constantibus, dissotiato per divortium mortis auriga qui quattuor istis partibus loca dederat, ne quisquam mobilitatem sui vehementius tolleret, dominante flagro, divisis in unum concordanti iunctura finibus, agitabat; hae, inquam, quottuor partes, recedente anima, in globum corporis viduati, excusso societatis auctore, miscentur".

12    Free translation cf. "Quantus illic, rogo vos, populi festinatus? Quae spectantium turba? Qualis tanti miraculi potuit esse concentos?".

13    Free translation cf. "Et post quadriduum Lazari lingua movetur, manus officio praeparantur, oculi suis in orbibus currunt, vestigia gressibus explicantur, auribus renovatur auditus, acies dirigitur in parentes. Cognatio redivivis obtutibus numeratur, vox prosapiae currit in auribus".

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
