# Peer review of "The Meaning of ‘Spiritual’ as Integral Health: From Hippocrates of Kos to the Potamius of Lisbon"

_religions, doi:10.3390/rel13090848_

Round 1

Reviewer 1 Report

Well done.

I caught a couple "typos" but a good presentation of a thorough study.

  The topics of Spirituality and Religion-&-Health are both very significant at this time and of great interest. This article combines them in a thorough historical approach that includes the development of medical science, Church history, Christian biblical scholarship and the evolving concept of spirituality. It connects with research and scholarship in several different fields, and I would think that it is of great interest to many Religions readers. While focused, it is inclusive of interdisciplinary scholarship and is quite wide-ranging. And it is well written.  
    “typos” ?  

      I am not an editor and don’t enjoy proof-reading, but I do notice when syntax, usage or a mistake break into the rhythm of my reading.  
      For example, at the beginning of the article, on page 2, second and third lines… “Hippocrates himself is an asclepiad, a kind of religious community of physician priests.”   Jarring.  

And in “7. Conclusion” second paragraph last sentence… “other ancient Christian authors, which…”  I always use “who” in that phrasing. 

Author Response

Reviewer 1

We are very grateful for the reading made, the observations made about the relevance of the text and we appreciate the feedback given, which certainly contributes to the quality of the text. We tried to review the text taking into account the issue of writing fluency, as well as correcting the questions on pages 2 and 7.

Kind regards

Reviewer 2 Report

The article offers an excellent engagement in primary sources with a clear analytical argument. More engagement with the secondary literature could help expand the audience/readership so the relevance of this work is readily understood. For ex. F. Retief/L. Clliers, "Influences of Christianity on Greco-Roman Medicine,"in Acta Theologica 2005, 259-77; R. Numbes and D. Amundsen, Caring and Curing: Health and Medicine....[although this is more of a introductory textbook presentation of related material]; D. Amundsen, Medicine, Society, and Faith in the Ancient and Medieval Worlds (1995). 

Overall this is a fascinating examination of the material. 

Author Response

Reviewer 2

We are very grateful for the reading made, the observations made about the relevance of the text and we appreciate the references suggested, that surely enriched the text, and who were all incorporated, excepted the book "Caring and Curing" which we also understand to be an introductory approach.

Due to the vacation times on Europe, it wasn't possible to use the volume of Amundsen (1995) in our University Library. We even ordered a copy, but the delivery time was longer than the deadline given to us by the publisher to make the corrections.

Otherwise, we used an earlier work of him with the same subject, namely: Amundsen, D. W. (1982). Medicine and Faith in Early Christianity. Bulletin of History of Medicine, vol. 56, p. 326-350.

Kind regards

Reviewer 3 Report

The topic of this paper is relevant and of high interest not only in historical perspective and can help to improve the understanding of the influence and reception of hippocratic medicine and terms in Christian understanding incl Christian self-understanding as a religion of healing (von Harnack), including the topic of Christus medicus.

In my opinion, the paper with its central scopes can be developed more coherently and more clearly. This applies especially to the reception and interpretation of Potamius of Lisbon's writings and understandings. Actually, in this regard the paper oscillates between Potamius' rhetorics and Potamius' theological anthropology and the role of Hippocratic anatomy (!) for both; this ought to be better distinguished and clarified.

The conclusion (ch. 7) then goes much beyond what has been discussed before and uses new sources (like the letter to Diognet) for a more complete vision (which I would welcome, too), although some seminal remarks are present in the previous chapters.

All in all, this paper deserves more logic coherence and a clearer development of the arguments provided more or less explicitly to make it a valuable contribution to theological discourse and to the discussion of the development of ideas.

I would also like to add a few elements for further improvement in detail:

1. The Abstract would better give, in addition to a clear idea of the scope of this paper, a short overview of the method and proceeding of this paper. Otherwise, please add an overview of your subsequent chapters making transparent the logic of your proceeding and development of your argument at the beginning of the introduction.

2. There is a mix of sources and literature used. Noteworthy publications are missing, including German collections on the topic of "Religion and Disease" (e.g. Etzelmüller/ Weissenrieder 2010). Please give a rationale for your selection of literature and of editions you use (e.g. p. 6 why do you provide the source in Italian? „Omelie sull’esamerone“).

3. It seems you express your central claim/ thesis at the end of the introduction: the Hippocratic use of the term [scil. Pneuma] is fundamental to determining the exact meaning of its incorporation into Greco-Roman philosophical and Christian theological thought. You could mark this more explicitly and that you are going to develop this thesis in the following chapters (if this is really what you intend to do). 

4. In chapter 5 there is an only subchapter 5.1, while one would expect that at least 5.2 would follow. As this does not follow, there is no need of 5.1. Please find another solution.

5. p. 8:“stoica salus, which for Christianity is an intrinsic part of soteriology“ - this identification is noted for the first time here without any proof and only in a relative clause: this deserves and needs better introduction and elaboration

6. Christus medicus and gratuitous healing action: Both are also including non-religious medical art/ treatment (“desacralized“) not in terms of Christianity as „health religion“ - (which could be rather characterized a religion of healing, as von Harnack had put it). Cf. p. 9: In the same sentence, we have two claims: the secularity of the medical art and the bodily integrity of the Christian salus.

This could be discussed more lucidly in order to show the reception of "rational medicine" in Christianity.

7. Cf. p. 10 - three points to clarify:

"It is significant how the taste for Potamian anatomy brings it closer to the human anatomy of Alexandrians or empiricists. (Nutton 144-156)". - Please explain what you mean by “taste for Potamian anatomy“ here.

Strange logic: “in the wake of an already long tradition of using knowledge in the field of medical science,“ - in which sense is a 4th century author like Potamius in the wake of an already long tradition ??- at least, this sounds contradictory.

“a Christian tradition receptive to the contribution of medical science to Christian theology“ - but it is also a reception of medical science as medical science (not only as imagery for theological notions): this is vital for Christian understandings of medicine and health

8. p. 11 and 12:

p. 11: grammatically, it must read “vultus hominis“, not “homini“

p. 11-12: better proportion of the two columns required in the layout

9. Not initatio but rather imitatio Christi (towards bottom p. 13)

10. p. 14:

"This represents a medical pedagogy that fits within the Christian medical pastoral between the centuries 3rd and 4th, that has been produced between Hippocratism and Christianity since the beginning of the Empire (León 519-521)." - Why do you refer to this here as something already known and why not much earlier???

Vis‘ rhetorical: ought to be rhetorical vis or rhetorical power.

11. p. 14:

Potamius’ originality resides in the vis’ rhetorical he attributes to medical art for the elaboration of his Christian anthropology and dogmatics.

This is important for the (lacking) coherence of the argument: Potamius is using medical art for rhetorical reasons, not (or much less so) for theological reasons or for a notion of holistic health - see next quote on Potamius‘ fascination… in 6.4.: His fascination with the body is such that the other dimensions of man seem forgotten.

12. pp. 15 and 16:

p. 15, towards bottom: memberrum natura??? - please check

p. 16 §3, towards the end of §: "Potamius, God the Father" - what does this mean?

13.

p. 17:

"The resurrection ends up being presented as a miraculous cure that resulted in the recovery of all bodily faculties: mouth; ears, eyes… Christ is the distributor of salutary remedies (salutaria)." - no reference to the source is given: is this explanation the author’s logic or of the source?

"The Potamian resurrection narrative emphasizes the close link between health and society, or between the individual body of Lazarus that is returned to the social body, represented by the sisters, friends, and the multitude of people who witness the miracle there and receive it. Returning to a healthy life coincides with a return to society and corresponds to the notion of integral health in Christianity in the first centuries." -

Having read the paper up to this point, this conclusion seems an over-interpretation very much which is not really shown convincingly before - beautiful or nice as it may be.

14. ch. 7:

The conclusion is loosely connected to the paragraphs before. It provides an understanding for which the exposition on Potamius is neither necessary nor really illustrative.

e.g.p. 18: "spirituality as a search for meaning, religious or not, is fundamental for integral health."

Modern as it may seem, is this true to Potamius? Beautiful affirmations at the end, yet not really conclusively covered by the previous chapters. Rather, they seem to be interpreted into the ancient texts.

15. Please check a few names and attributions:

e.g. p. 7 “Jewish philosopher Alexandrian“ should read “the Jewish Alexandrian philosopher“, refering to Philo; and “Irineu de Lião“ should be “Irenaeus of Lyon“ as well as “Minúcio Félix“ is “Minucius Felix“, p. 9 “Máximo Confessor“ is Maximus Confessor; please also check the use of other names and unify the use. 

16. There is an affirmation which I deem central on p. 17 towards the bottom:

"Here the religious question should not be confused with the therapeutic one, but a possibility of a positive reading would reside in the fact that an authentic religious experience, while seeing in it a deep meaning of life, has therapeutic effects. The same goes for the therapy that, being positive, expands the possibilities of a healthy religious experience."

You interpret and present Potamius as "kind of antidote to the tendency towards spiritualization of the pneuma, to accentuate its aspect of integrating, vitalizing and unifying pneuma dynamism." (in a medical sense?)

This and hopefully my indications could serve well to revise the paper and to better focus it with the contribution by Potamius you think worthwhile for the topic...

Author Response

Firstly, we really appreciate the careful reading and the very relevant observations that made it possible to think of ways to improve the text. We seek to value the evaluation work done and try to answer all the questions raised. We remain at your disposal for any other questions and suggestions that may still be indicated. Our sincere thanks!

Round 2

Reviewer 3 Report

A humble "thank you" for taking into account my indications so thoroughly for your paper and your further development of it.